# Differential impact of a dyskeratosis congenita mutation in TPP1 on mouse hematopoiesis and germline

Jacqueline V Graniel[1,2,3], Kamlesh Bisht[1,4], Ann Friedman[5], James White[6,7], Eric Perkey[2,8,9], Ashley Vanderbeck[9], Alina Moroz[7], Léolène J Carrington[9], Joshua D Brandstadter[9], Frederick Allen[9], Adrienne Niederriter Shami[2,6], Peedikayil Thomas[6,7], Aniela Crayton[7], Mariel Manzor[7], Anna Mychalowych[7], Jennifer Chase[5,8], Saher S Hammoud[6], Catherine E Keegan[6,7], Ivan Maillard[9], Jayakrishnan Nandakumar[1]

**Telomerase extends chromosome ends in somatic and germline stem cells to ensure continued proliferation. Mutations in genes critical for telomerase function result in telomeropathies such as dyskeratosis congenita, frequently resulting in spontaneous bone marrow failure. A dyskeratosis congenita mutation in TPP1 (K170Δ) that specifically compromises telomerase recruitment to telomeres is a valuable tool to evaluate telomerase-dependent telomere length maintenance in mice. We used CRISPR-Cas9 to generate a mouse knocked in for the equivalent of the TPP1 K170Δ mutation (TPP1 K82Δ) and investigated both its hematopoietic and germline compartments in unprecedented detail. TPP1 K82Δ caused progressive telomere erosion with increasing generation number but did not induce steady-state hematopoietic defects. Strikingly, K82Δ caused mouse infertility, consistent with gross morphological defects in the testis and sperm, the appearance of dysfunctional seminiferous tubules, and a decrease in germ cells. Intriguingly, both TPP1 K82Δ mice and previously characterized telomerase knockout mice show no spontaneous bone marrow failure but rather succumb to infertility at steady-state. We speculate that telomere length maintenance contributes differently to the evolutionary fitness of humans and mice.**

## Introduction

Telomeres are nucleoprotein complexes that make up the natural ends of eukaryotic chromosomes. They consist of tandem, hexameric DNA repeat sequences (GGTTAG in mammals) that are mostly double-stranded (10–15 kb in humans) and end in a short single-stranded (ss) G-rich overhang (50–500 nt in humans) (Palm & de Lange, 2008). Telomeric DNA is bound by a six-protein complex called shelterin, which protects chromosome ends from participating in unwanted end-to-end fusion/degradation events (Palm et al, 2009). The inability of DNA polymerases to replicate the 5′ end of the lagging strand would result in the gradual shortening of telomeric DNA with every round of cell division (Levy et al, 1992). Whereas progressive telomere shortening is warranted in our somatic cells, as it can help prevent unregulated cell division associated with cancer, the end replication problem must be countered in long-lived proliferating cells such as germline and somatic stem cells (Shay & Wright, 2010; Pech et al, 2015). Telomerase is a unique ribonucleoprotein reverse transcriptase that helps solve the end replication problem by synthesizing new telomeric DNA repeats at the ends of chromosomes using an internal RNA template (Greider & Blackburn, 1985, 1989; Lingner et al, 1997; Meyerson et al, 1997).

Mutations that compromise telomere length maintenance result in diseases termed telomeropathies, the most prominent example of which is dyskeratosis congenita (DC) (Dokal, 2011; Armanios & Blackburn, 2012; Niewisch & Savage, 2019). Severe shortening of telomeres in patients with DC eventually results in BM failure, which is the most common cause of death (Ballew & Savage, 2013; Collins & Dokal, 2015). Hematopoietic stem cells (HSCs) and hematopoietic progenitors derived from DC patients exhibit reduced self-renewal, providing a cellular basis for BM failure (Jones et al, 2016). DC can present with a broad phenotypic spectrum, including a diagnostic triad of epithelial manifestations (dysplastic nails, abnormal skin pigmentation, and oral leukoplakia), strongly indicative of somatic stem cell failure (Savage, 2014). DC and other telomeropathies display an earlier onset and worse prognosis of the disease in later generations, a phenomenon known as genetic anticipation (Savage & Bertuch, 2010). Genetic anticipation in these diseases results from progressive shortening of telomeres caused by inheritance of both

[1]Department of Molecular, Cellular and Developmental Biology, University of Michigan, Ann Arbor, MI, USA   [2]Medical Scientist Training Program, University of Michigan, Ann Arbor, MI, USA   [3]Department of Cell and Developmental Biology, University of Michigan, Ann Arbor, MI, USA   [4]Oncology Therapeutic Area, Sanofi, Cambridge, MA, USA   [5]Department of Internal Medicine, Michigan Medicine, Ann Arbor, MI, USA   [6]Department of Human Genetics, University of Michigan, Ann Arbor, MI, USA   [7]Department of Pediatrics, Michigan Medicine, Ann Arbor, MI, USA   [8]Cellular and Molecular Biology Program, University of Michigan, Ann Arbor, MI, USA   [9]Division of Hematology/Oncology, Department of Medicine; Abramson Family Cancer Research Institute, University of Pennsylvania, Perelman School of Medicine, Philadelphia, PA, USA

Correspondence: jknanda@umich.edu; imaillar@pennmedicine.upenn.edu; keeganc@med.umich.edu

short telomeres and the causative mutation from the affected parent's gamete. Consistent with shortened telomeres playing a causal role in somatic stem cell failure in DC, most of the 14 genes found mutated in DC encode either a subunit of the telomerase holoenzyme or a factor directly involved in telomerase biogenesis, trafficking, or recruitment to the telomere (Grill & Nandakumar, 2020).

The protein ACD/TPP1 (adrenocortical dysplasia homolog/TINT1-PTOP-PIP1, hereafter mentioned as TPP1; human gene name: ACD; mouse gene name: Acd; HUGO Gene Nomenclature Committee Symbol: ACD) is the only shelterin component to date involved in both chromosome end protection and end replication (Houghtaling et al, 2004; Liu et al, 2004; Ye et al, 2004). TPP1 performs its distinct functions using different protein domains. C-terminal protein–protein interaction domains facilitate TPP1's interactions within shelterin (Palm & de Lange, 2008; Chen et al, 2017; Hu et al, 2017; Rice et al, 2017; Grill et al, 2021). TPP1 recruits the ss telomeric DNA-binding protein POT1 to prevent illicit homologous recombination of telomeric overhangs (Guo et al, 2007; Hockemeyer et al, 2007). A hypomorphic allele of Acd (acd/acd) results in complex developmental defects, which include severe growth retardation, hyperpigmentation, and urogenital defects. These findings are consistent with the disruption of the end protection function of TPP1, which would be expected to unleash genome instability (Keegan et al, 2005). In contrast, the N-terminal oligonucleotide/oligosaccharide-binding (OB) domain of TPP1 is responsible for associating with the catalytic subunit of telomerase called TERT (Telomerase Reverse Transcriptase) (Fig 1A) (Xin et al, 2007; Zhong et al, 2012). By recruiting telomerase to the telomere, TPP1 directly facilitates end replication (Wang et al, 2007; Xin et al, 2007; Abreu et al, 2010; Nandakumar & Cech, 2013).

A combination of mutagenesis screens and inter-species domain swap experiments revealed two key locations within TPP1 that are critical for telomerase recruitment: the TEL patch (TPP1 glutamate [E] and leucine [L]-rich patch) and the NOB (N terminus of the OB domain) (Nandakumar et al, 2012; Sexton et al, 2012; Zhong et al, 2012; Bisht et al, 2016; Grill et al, 2018; Tesmer et al, 2019). The human TPP1 TEL patch consists of a highly conserved acidic loop [266]DWEEKE[271] that is critical for telomerase processivity, recruitment to telomeres, and telomere length maintenance in cultured human cells (Fig 1B) (Nandakumar et al, 2012). Within this loop is a single basic amino acid K170 that was found to be deleted (K170Δ) in two separate families with telomeropathies (Guo et al, 2014; Kocak et al, 2014). Individuals carrying a heterozygous K170Δ mutation displayed short telomeres, with one proband suffering from BM failure and a severe form of DC known as Hoyeraal–Hreidarsson syndrome (Kocak et al, 2014). Structural analysis of TPP1 OB containing K170Δ suggests that the K170 residue ensures proper orientation of the acidic residues in the TEL patch loop to facilitate TERT binding (Bisht et al, 2016). TPP1 K170Δ abrogates telomerase recruitment to telomeres and reduces the ability of TPP1 to stimulate telomerase processivity in vitro (Kocak et al, 2014). However, consistent with this mutation being outside of TPP1's end protection domains, it does not impact binding to POT1 or the protection of telomeres in vivo (Kocak et al, 2014; Bisht et al, 2016; Grill et al, 2021). CRISPR-Cas9–mediated knock-in of one allele of K170Δ into HEK 293T cells retaining one wild-type (WT) allele resulted in a progressive

shortening of telomeres with population doubling, suggesting that this one amino acid deletion in TPP1 protein is sufficient to shorten telomeres in human cells (Bisht et al, 2016).

Pioneering work using mouse models of telomerase-deficient mice has tested the importance of telomere maintenance in mammals. Complete KO of the mouse telomerase RNA subunit ($mTR^{-/-}$) resulted in a decrease in telomere length in successive generations, as expected from genetic anticipation (Blasco et al, 1997; Herrera et al, 1999). Surprisingly, the mice did not develop any overt BM failure even after six generations (Lee et al, 1998). Unlike patients with DC, the $mTR^{-/-}$ mice showed no uniform hematopoietic failure through complete blood count (CBC) or altered cellular composition in the spleen and BM by flow cytometry under steady-state conditions (Lee et al, 1998; Rudolph et al, 1999). However, under conditions of stress, such as caused by serial BM transplantation and/or severely short starting telomere length, hematopoietic defects did appear in $mTR^{-/-}$ mice (Samper et al, 2002; Allsopp et al, 2003; Choudhury et al, 2007; Rossi et al, 2007; Armanios et al, 2009; Sekulovic et al, 2011). Although some hair graying or alopecia occurred in later generation $mTR^{-/-}$ mice, the diagnostic triad of cutaneous DC symptoms was also absent (Rudolph et al, 1999). DC-like features appeared in genetic backgrounds combining telomerase KO with Pot1b KO (Hockemeyer et al, 2008) or in strains of laboratory-derived inbred mice with short telomeres (e.g., CAST/EiJ) (Armanios et al, 2009), but not in the wild-type laboratory strain. Despite a clear lack of spontaneous BM failure in $mTR^{-/-}$ mice under steady-state, a conspicuous infertility phenotype emerged in late-generation $mTR^{-/-}$ mice (Lee et al, 1998). Fertility diminished in both males and females, and deeper analysis revealed morphological defects in male and female reproductive organs and apoptotic clearance of germline cells in the early stages of sperm development (Lee et al, 1998; Hemann et al, 2001).

The tissue-specific phenotypic manifestations in telomerase-deficient mice are reversed in DC patients (Keefe, 2016; Gansner et al, 2017; Giri et al, 2017, 2021). While the BM failure is overt, the connection with fertility is tenuous, barring isolated case studies that show evidence for reduced levels of Anti-Mullerian Hormone (marker for ovarian reserve) (Giri et al, 2017) or incidence of infertility in patients with other telomeropathies such as idiopathic pulmonary fibrosis (Alder et al, 2015). Thus, although telomere shortening and degeneration in reproductive tissues occurs during normal aging in humans (Keefe, 2016), the primary vulnerability in humans caused by telomere shortening seems to lie in the BM and other somatic tissues, not the germline. Despite the divergent manifestation of telomerase-dependent telomere length maintenance defects in $mTR^{-/-}$ mice and human patients suffering from telomeropathies, a detailed investigation of the developmental programs of hematopoiesis and gametogenesis, including in-depth analysis of stem, progenitor, and differentiating cell populations, has not been performed using equivalent genetic perturbations.

Telomerase deletion in mice does not recapitulate the somatic stem cell deficiency seen in DC patients and instead points to a selective disruption of germline stem cells in the absence of telomerase. However, it has not been established if the observed reproductive phenotypes are the result of complete KO of telomerase, a more severe genetic perturbation than mutations in DC, which are often more subtle in nature. DC mutations provide a

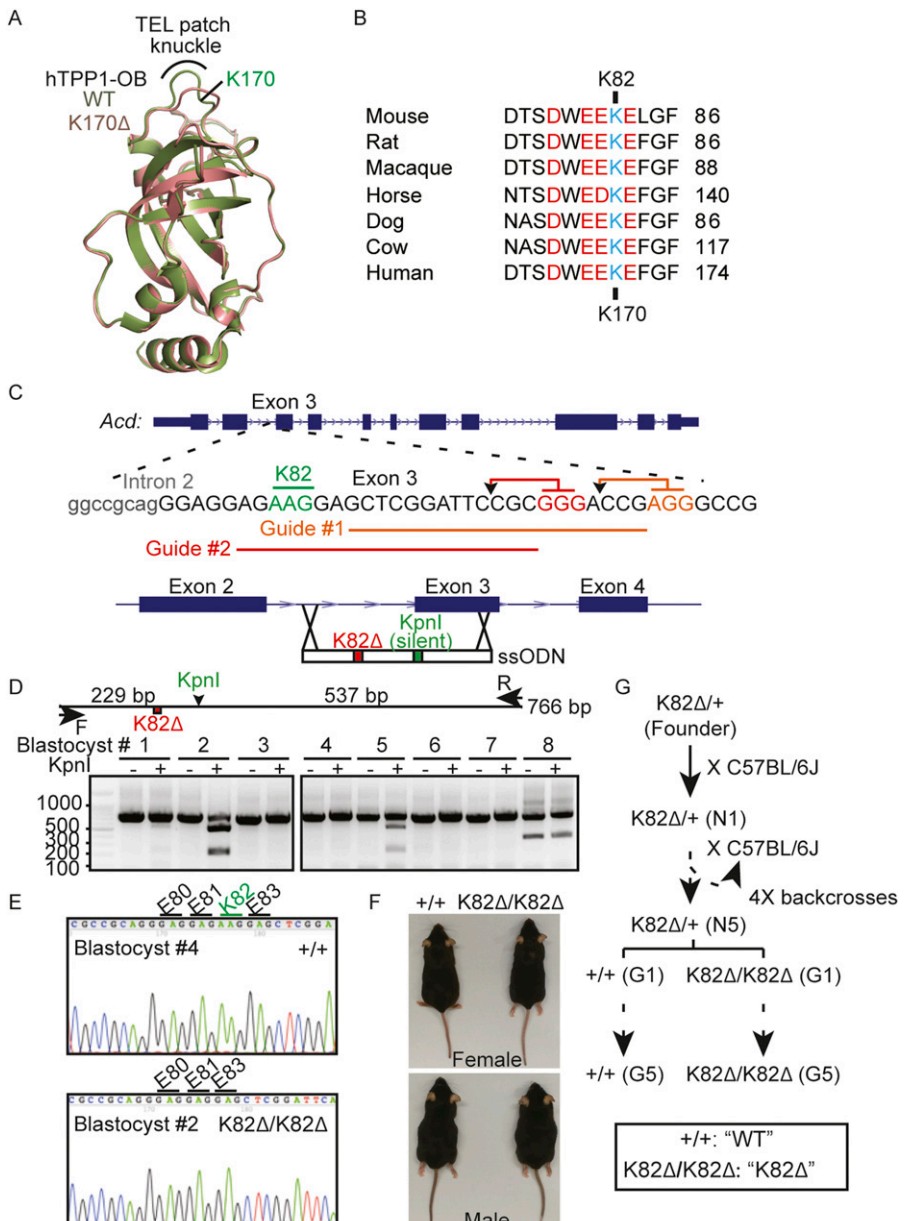

**Figure 1. CRISPR-Cas9 generation of a mouse model of a dyskeratosis congenita mutation in TPP1.**
**(A)** Overlay of the crystal structures of human TPP1 OB (hTPP1 OB) domains from WT and K170Δ proteins. The deletion of K170 distorts the acidic TEL patch knuckle. **(B)** Strict conservation of the TEL patch loop (acidic amino acids shown in red) that also harbors the human TPP1 K170 residue (shown in cyan) that is deleted in dyskeratosis congenita. The mouse equivalent of this residue is K82. **(C)** The PAM sites (GGG and AGG highlighted in red and orange, respectively) and Cas9 cleavage sites (arrowheads) for the two guide RNAs used to cleave the exon coding for mouse TPP1 K82Δ are shown. The schematic for the homologous recombination repair template shows not only the deletion of the K82 codon (red) but also a silent mutation that creates a KpnI site (green) for screening purposes. **(D)** PCR amplification and KpnI restriction digestion screening of eight blastocysts after injection with guide RNAs and repair template for introducing the TPP1 K82Δ mutation. Expected sizes for PCR amplicon and its KpnI cleavage products used to screen the editing of the *Acd* locus of mouse blastocysts by CRISPR-Cas9 are shown above the gel. **(D, E)** Sanger sequencing of the PCR products of the indicated blastocysts (same as those analyzed in panel D) showing accurate editing of the *Acd* locus. **(F)** Images of G1 WT and K82Δ (homozygous), male and female mice. **(G)** Breeding scheme to backcross the CRISPR-edited K82Δ/+ founder mouse and generate WT and homozygous K82Δ mice that were bred for five generations (G1 → G5).

physiologically relevant avenue to evaluate the importance of telomere length regulation in different tissues, including those of the soma and germline. DC mutations in the *TINF2* gene coding for shelterin protein TIN2 have been modeled in mice, but their underlying mechanism is complicated, with several proposed models relying both on telomerase-dependent and independent modes of action (Savage et al, 2008; Walne et al, 2008; Canudas et al, 2011; Yang et al, 2011; Frescas & de Lange, 2014; Frank et al, 2015; Nelson et al, 2018; Grill & Nandakumar, 2020). The wealth of data surrounding TPP1 K170Δ that directly link the disruption of protein structure and function caused by this mutation to telomere shortening makes it a valuable tool to investigate telomerase-mediated telomere length maintenance in the mouse soma and germline. Here, we introduced the equivalent of human TPP1 mutation K170Δ in mice,

TPP1 K82Δ, and performed a detailed investigation of hematopoietic and germline development. This is the first mouse model to evaluate the importance of telomerase-mediated telomere lengthening without perturbing end protection, or the telomerase holoenzyme. Our study also marks, to our knowledge, the most in-depth analysis of the stem cell compartments involved in both hematopoiesis and reproduction in a mouse model of a DC mutation. We observed that TPP1 K82Δ shortened telomeres with successive generations. However, K82Δ did not lead to BM failure or other defects in the hematopoietic hierarchy. In stark contrast, K82Δ mice showed reproductive defects that ultimately resulted in sterility in late generations, suggesting that telomere shortening induced by a single amino acid deletion in TPP1 is sufficient to elicit a mouse germline defect and infertility. Our studies support a

model wherein the mouse germline, but not the mouse BM, is especially vulnerable towards defects in telomerase-dependent telomere length maintenance, having important implications for the differential vulnerabilities of mice and humans towards telomere shortening.

# Results

## Generation of viable mice homozygous for the DC mutation TPP1 K82Δ/K82Δ via CRISPR-Cas9 knock-in

Using CRISPR-Cas9 technology in the presence of a ss oligo donor (ssODN) containing the K82Δ mutation as a template for homologous recombination, we successfully edited the mouse *Acd* genomic locus to generate a mouse heterozygous for TPP1 K82Δ (Fig 1A–G). After five generations of backcrossing of this heterozygous founder to the C57BL/6 inbred background (N1 → N5), we intercrossed the resulting progeny to obtain generation 1 (G1) TPP1 K82Δ/K82Δ homozygous mice (referred to as K82Δ hereafter) and ACD/TPP1 +/+ age-matched mice (referred to as WT hereafter) as controls (Fig 1G). Although K170Δ is heterozygous in human DC, we decided to study the mutation in a homozygous context to accelerate phenotypic progression in mice. This was important as laboratory mice have significantly longer telomeres than humans (Kipling & Cooke, 1990), making it unlikely for phenotypes to appear in a heterozygous context or over a few generations. Accordingly, we intercrossed G1 K82Δ (and in parallel, G1 WT) mice to generate successive generations of mice (G2, G3, G4, and G5) using a breeding scheme closely resembling that used for the generation of *mTR*⁻/⁻ mice (Fig 1G) (Blasco et al, 1997; Herrera et al, 1999).

## The TPP1 K82Δ mutation causes progressive telomere shortening across generations

We adapted Flow-FISH, a highly quantitative approach used routinely in the clinical setting, to measure the length of telomeres in WT and K82Δ BM cells (Baerlocher et al, 2006; Sekulovic et al, 2011). Flow cytometry coupled with Flow-FISH combines the power of high-speed single-cell analysis to evaluate distinct populations of cells with FISH to visualize telomeric DNA and measure telomere length in these cells. To optimize Flow-FISH for mouse cells and provide internal calibration controls, we used calf thymocytes, which are phenotypically distinct and contain much shorter telomeres than mouse BM cells. We hybridized a fluorescently labeled telomeric peptide nucleic acid (PNA) probe to telomeres and used LDS-751 to stain DNA in the nucleus of mouse BM cells from mutant and WT mice. Quantitation of calf thymocyte telomere length using Southern blot analysis was used as a calibration step to calculate the length of mouse BM telomeres (Fig S1A–I). Dot plot visualization of the fluorescence intensity in the PNA and LDS-751 channels in the presence and absence of fluorescent telomere PNA probe confirmed the distinct flow cytometric features and telomere length signals of mouse BM cells versus calf thymocytes (Fig 2A–C). Histograms derived from the flow cytometry plots revealed that G1 WT BM had a mean telomere length of 38.1 kb, whereas G1 K82Δ had significantly shorter telomeres (mean = 33.3 kb) (Fig 2D and E). Telomeres in K82Δ BM cells shortened progressively with

generation number, with G5 K82Δ BM exhibiting a mean telomere length of 25.7 kb (Figs 2F and G and S1J). These findings, which reveal a ~2 kb shortening of telomeres per generation in the K82Δ BM, are consistent with observations made with the equivalent human mutation, which is associated with severely shortened telomeres in affected individuals and causes telomere shortening in cultured cells (Guo et al, 2014; Kocak et al, 2014; Bisht et al, 2016).

## K82Δ mutant mice do not develop BM failure up to at least five generations under steady state

Given that BM failure is the primary cause of morbidity in DC patients, including in the proband harboring TPP1 K170Δ (Kocak et al, 2014), we evaluated the hematopoietic system of WT and K82Δ mice across successive generations. CBCs were measured at multiple time points in G1 and G3 mice (Fig S2A–E). We did not observe any consistent differences between WT and mutant CBCs in either sex, within generations, or across generations, that would be reflective of BM failure in the mutant mice. Specifically for the white blood cell count (WBC), most time points did not reveal significant differences in female K82Δ mice except for a decreased WBC in G3 mice at 7 mo. In males, an apparent decreased WBC was observed in G1 16-mo-old mice, but it was not sustained at 2 yr of age. Male G3 K82Δ mice had a decreased WBC at 7 mo that did not recapitulate at 12 or 16 mo of age (Fig S2A). Both male and female K82Δ mice had no changes in platelets, red blood cells, or hemoglobin levels (Fig S2B–D). Male but not female K82Δ mice had a mildly increased mean corpuscular volume compared with WT (Fig S2E). The lack of gross changes in CBC with K82Δ is consistent with previous studies with *mTR*⁻/⁻ mice (Lee et al, 1998).

Peripheral blood measurements in steady-state conditions do not fully reflect hematopoietic stem and progenitor cell health in the BM. For a more comprehensive investigation of the hematopoietic system, we performed terminal harvests of G3, G4, and G5 mice, and probed stem, progenitor, and mature cell populations in the following hematopoietic and lymphoid organs: BM, spleen, and thymus (Figs S3A–O, 4A–G, and S5A–G). Even at the latest generations, G4 and G5 K82Δ mice showed preserved BM cellularity compared with WT mice (Fig 3A and B). Flow cytometric analysis showed a preserved frequency of lineage negative (Lin⁻) cells (Fig 3C–E), Lin⁻Sca-1⁺c-Kit^high (LSK) cells (Fig 3F–H), and CD150⁺ CD48⁻ LSK cells, which are highly enriched for long-term (LT)-HSCs, in G5 K82Δ mice compared with WT (Fig 3I–K). When we quantified additional progenitor and mature cell subsets in the BM, there was no consistent change in K82Δ mice compared with WT, although selected sex-related differences were apparent (Fig S3A–O). Similarly, we observed no changes in splenic cell populations that persisted over generations in K82Δ mice or were shared between males and females (Fig S4A–G). The same was true for early T lineage progenitors and successive downstream populations of developing thymocytes (Fig S5A–G). Given these observations, we conclude that the presence of a TPP1 K82Δ mutation does not result in spontaneous BM failure in mice up to at least five generations. In agreement with our results in the BM, we did not observe skin aberrations, including fur loss or hyperpigmentation, indicating that K82Δ did not have clinically apparent effects on epidermal cell homeostasis up to G5 (data not shown). Our results, combined with the rich literature surrounding *mTR*⁻/⁻ mice, strongly suggest a distinct resilience of

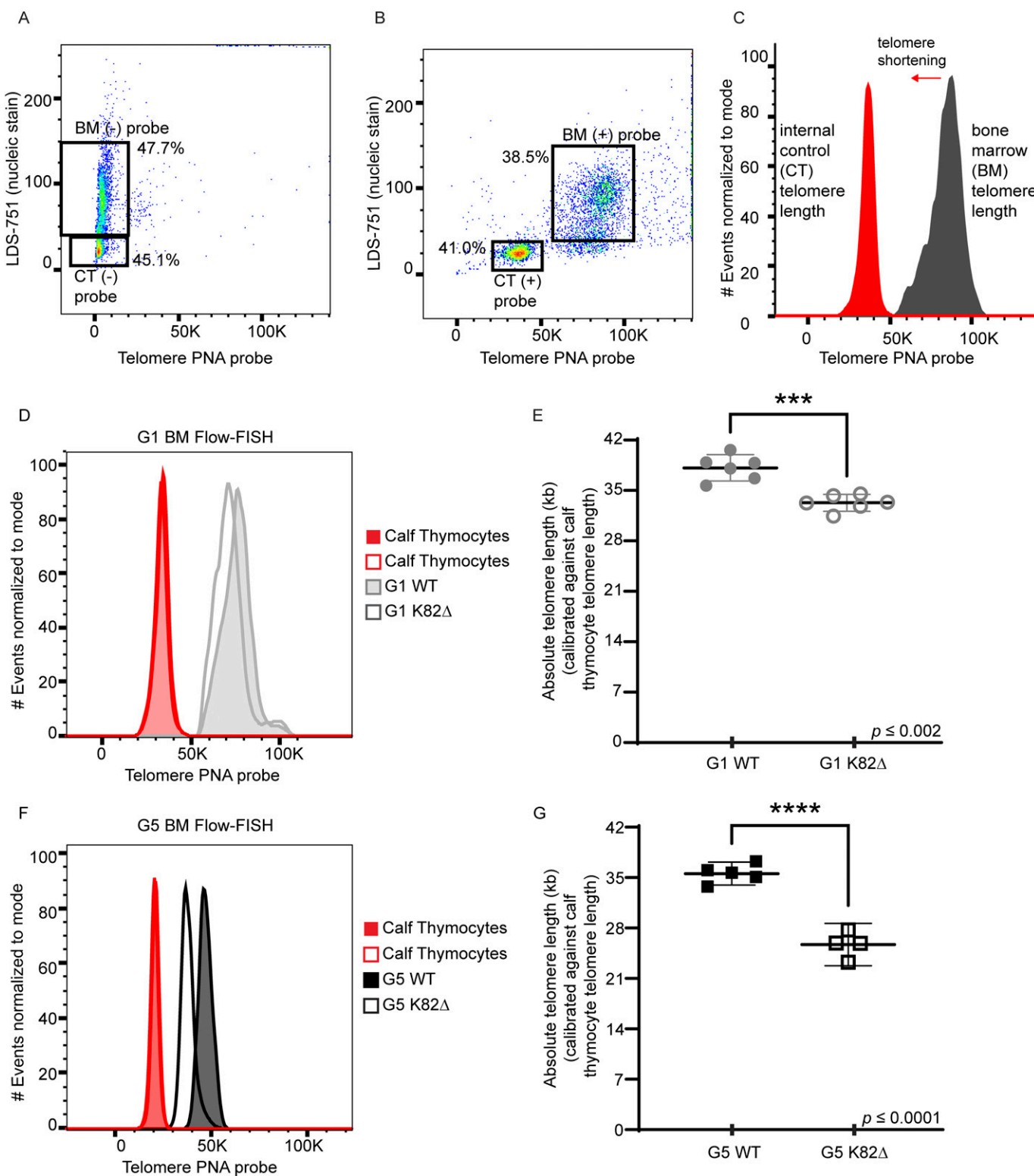

**Figure 2.  Flow-FISH to measure telomere length reveals progressive telomere shortening in K82Δ mutant mice.**
**(A, B)** Representative flow cytometry plots of Flow-FISH of WT G1 mouse samples with (A) no probe and (B) PNA telomere probe. Gating includes calf thymocytes (+/− probe) and BM (+/− probe). **(C)** The histogram shows data with the probe. Red indicates calf thymocyte telomeric probe signal. Grey indicates BM telomeric probe signal. **(D)** Histogram showing telomeric probe signal of G1 WT (grey filled peak) and K82Δ (grey open peak) alongside their internal calf thymocyte controls used in the same experiment. **(E)** Quantitation of absolute telomere length of G1 WT and K82Δ calibrated against calf thymocytes telomere length obtained by TRF analysis. **(F)** Histogram showing telomeric probe signal of G5 WT (black filled peak) and K82Δ (black open peak) alongside their calf thymocyte controls used in the same experiment. **(G)** Quantitation of absolute telomere length of G5 WT and K82Δ calibrated against calf thymocytes telomere length obtained by TRF analysis. n = 4–6 mice per condition; mean with 95% CI; significance calculated with Prism software using t test for individual experiments; ***P ≤ 0.002, ****P ≤ 0.0001.

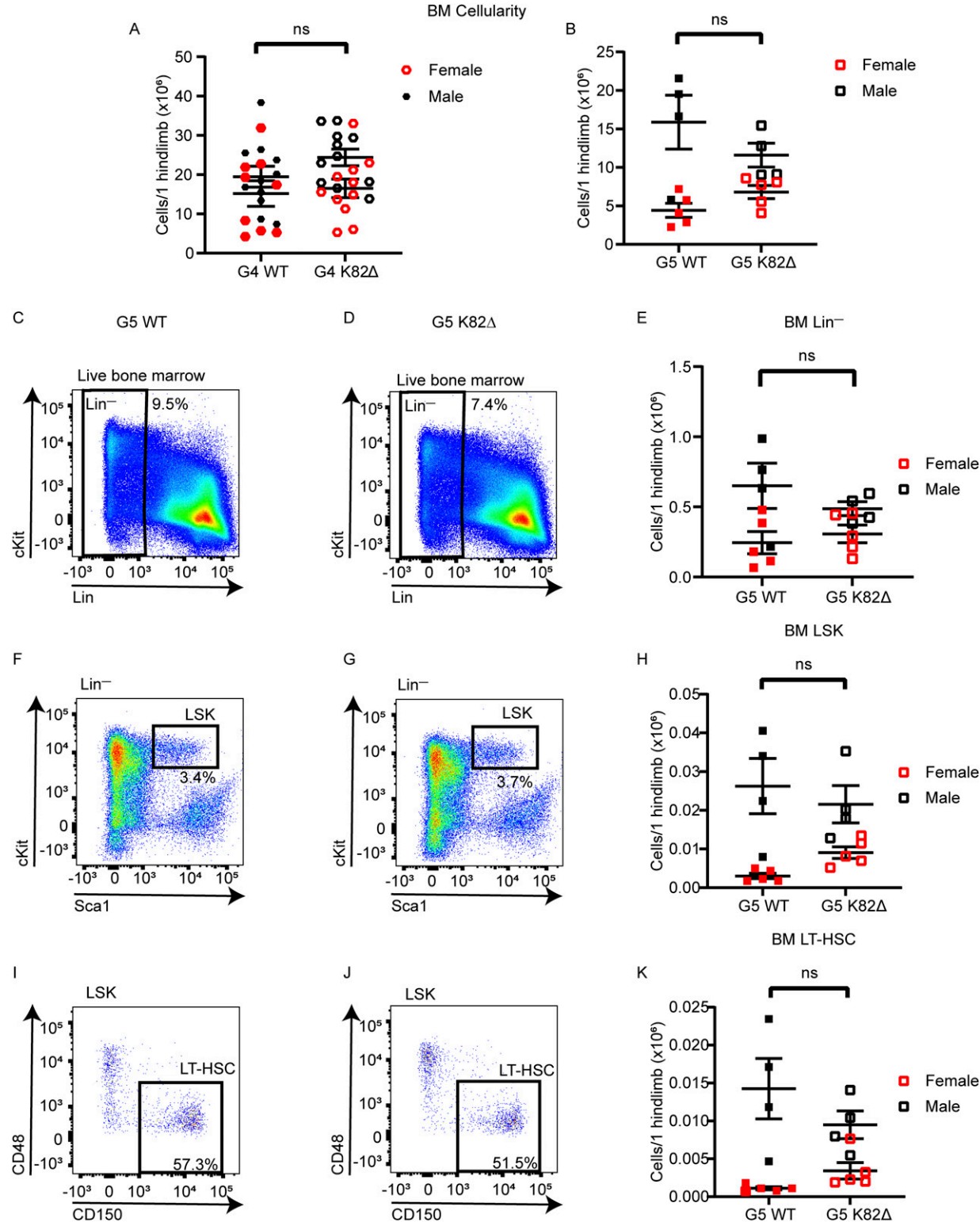

**Figure 3. K82Δ mutant mice do not develop BM failure.**
**(A, B)** BM cellularity for 1 hindlimb per mouse for WT and K82Δ mice in (A) G4 and (B) G5. **(C, D, E, F, G, H, I, J, K)** Representative (C, D) lineage negative (Lin⁻), (F, G) Lin⁻Sca-1⁺c-Kit^high (LSK), and (I, J) CD150⁺ CD48⁻ LSK cells (LT-HSCs) flow cytometry plots of G5 WT (C, F, I), G5 K82Δ mice (D, G, J) with quantitation (E, H, K) showing equal frequencies in K82Δ mice compared with WT. Red indicates female mice, black indicates male mice, filled symbols indicate WT, and open symbols indicate K82Δ mice. n ≥ 4; mean with 95% confidence interval.

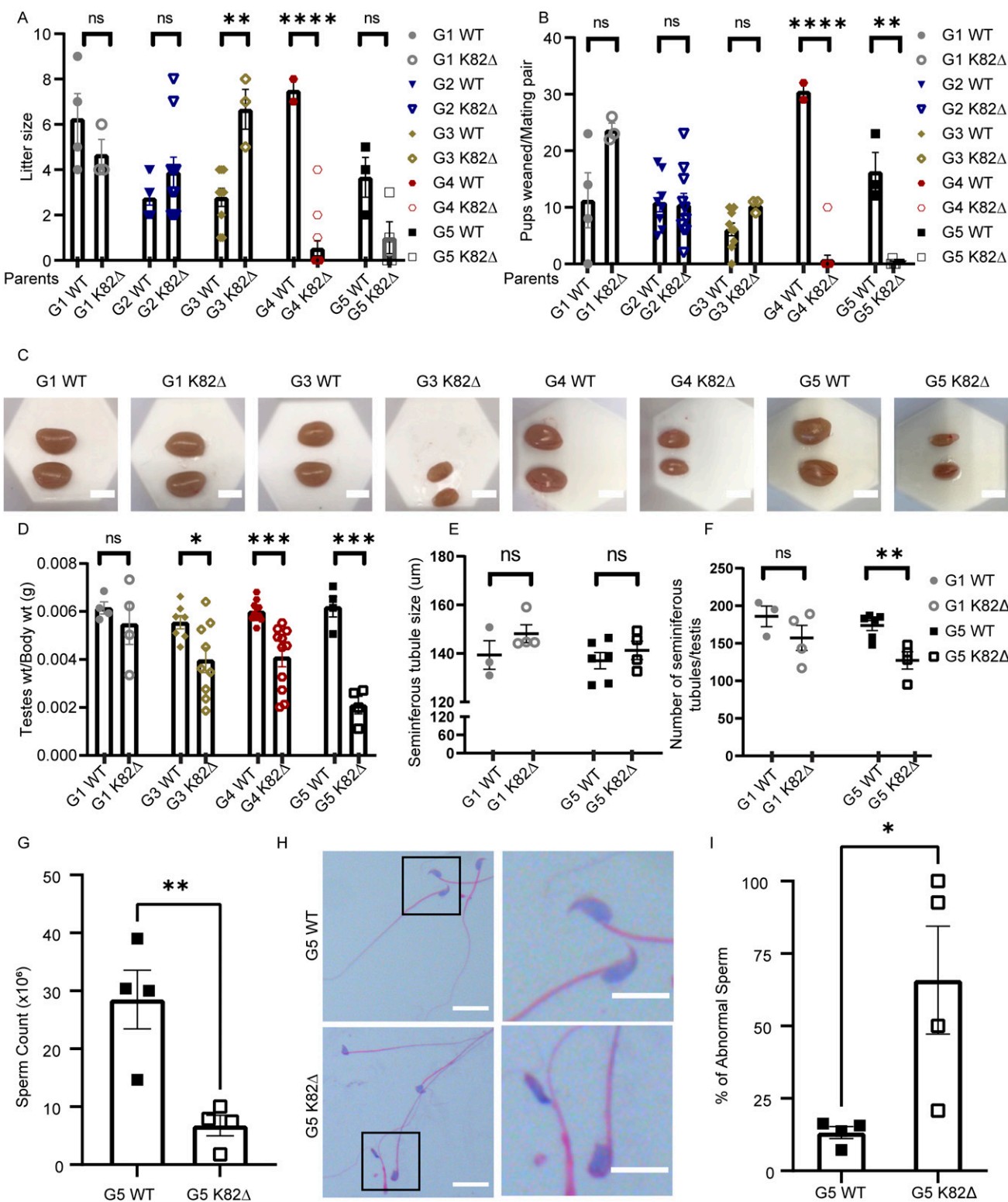

**Figure 4. TPP1 K82Δ mutation leads to reproductive defects in later generations.**
**(A, B)** X-axis indicates genotype of the parents. **(A)** Quantitation of the number of offspring per litter for the indicated generation number and genotype analyzed in this study.
**(B)** Quantitation of the total number of pups that survived past weaning per mating pair for the indicated generation number of WT and K82Δ mice. Number of mating pairs used for data in panels (A) and (B): G1 WT: n = 4; G1 K82Δ: n = 3; G2 WT: n = 8; G2 K82Δ: n = 10; G3 WT: n = 9; G3 K82Δ: n = 3; G4 WT: n = 2; G4: K82Δ n = 13; G5 WT: n = 3; G5 K82Δ: n = 4. **(C)** Representative gross morphology images of testes for the indicated generation number and genotype of mice studied. **(D)** Quantitation of testes/body weight of WT and K82Δ mice. n = 4–11. **(E, F)** Whole testis sections were stained with H&E and imaged and analyzed for seminiferous tubule (E) diameter and (F) number in G1 and G5, WT, and K82Δ, mice. At least two mice were evaluated per generation and genotype. Slides separated by at least 100 μm were used as technical replicates. **(G)** Number of sperm collected from vas deferens and

the mouse BM and other somatic components to telomerase-dependent telomere shortening as compared with DC patients.

## K82Δ results in reproductive defects and progressive loss of fertility

We next turned our attention to fertility as a readout for germ cell function, as complete KO of telomerase is known to cause infertility in late-generation mice (Blasco et al, 1997; Herrera et al, 1999). We observed a striking decrease in fertility in G4 K82Δ mice, as evidenced by a reduced number of litters per breeding pair, a trend that persisted in G5 K82Δ mice (Fig 4A). Remarkably, only one out of 13 G4 K82Δ breeding pairs produced litters. Among four breeding pairs set up for G5 K82Δ mice, only one female G6 K82Δ was born, thereby terminating the breeding of the K82Δ line. Furthermore, of the mice born, litter size and the total number of pups that survived past weaning was significantly lower in G4 and G5 K82Δ breeding pairs, suggesting that impaired germline proliferation and/or maturation caused by shortened telomeres resulted in prenatal and perinatal lethality of K82Δ mice (Fig 4B). Both sexes were affected as breeding of G5 K82Δ male and female mice to WT females and males, respectively, did not result in progeny (data not shown). The reproductive defects observed for TPP1 K82Δ are reminiscent of sterility observed in acd/acd hypomorphic mice (Keegan et al, 2005) and late-generation $mTR^{-/-}$ mice (Lee et al, 1998; Hemann et al, 2001). It is striking that deletion of a single amino acid in the TPP1 TEL patch region is sufficient to cause mouse infertility, demonstrating that disruption of telomerase-dependent telomere length maintenance causes mouse infertility in late generations under conditions where no spontaneous BM defects are evident.

## K82Δ mutation in mice reduces testis/body weight ratio and sperm counts

To further understand the basis for reduced fertility in K82Δ mice, we examined testicular size and observed a significant reduction in testis weight/body weight in G3–G5 male mutant mice (Fig 4C and D). Cross-sections of G5 K82Δ testes revealed no changes in the diameter of seminiferous tubules (Fig 4E), but a reduction in the number of tubules was observed compared with WT (Fig 4E and F). Testosterone levels remained unchanged (data not shown). Furthermore, G5 K82Δ male mice had lower sperm counts (Fig 4G) and a significantly increased proportion of sperm with abnormal morphology (Fig 4H and I). Specifically, the mutant sperm lacked the hook-like structure involved in sperm progression and attachment to the female reproductive tract (Varea-Sanchez et al, 2016). Together, these data indicate that TPP1 K82Δ mice have profound reproductive defects that could arise from underlying defects in the germline and/or somatic compartments of the testis.

## K82Δ results in an increased incidence of disordered and empty seminiferous tubules

We next analyzed seminiferous tubule cellularity to gain deeper insights into the reproductive defects observed in K82Δ mice.

Spermatogenesis involves three major developmental phases: spermatogonial proliferation, spermatocyte differentiation, and spermiogenesis. These developmental phases occur asynchronously in a radially organized manner, initiating at the basement membrane and moving toward the lumen as spermatogenesis proceeds (Fig S6A–E) (Larose et al, 2019). The germline stem cells reside along the basement membrane and undergo a series of mitotic divisions before differentiating to form primary spermatocytes. Primary spermatocytes enter meiosis I to produce secondary spermatocytes, which give rise to early spermatids. These spermatids undergo spermiogenesis, which is a 16-step process defined by the development of the acrosome from early spermatid (no acrosome, step 1) to premature spermatozoa (hooked acrosomes, step 16) (Nakata et al, 2015). To monitor acrosome dynamics and determine the rough stage of the seminiferous tubule cycle, we used PNA lectin immunofluorescence (Fig 5A) (Nakata et al, 2015). We collapsed the 12 stages of the seminiferous tubule into four bins (stages I–III; stages IV–VI; stages VII–VIII; stages IX–XII) and quantified the frequency of tubule stages in WT and mutant mice. For these studies, we used the hypomorphic acd/acd mouse model as a positive control for a severe germline defect as it is known to exhibit degenerated tubules (Keegan et al, 2005). No differences in spermatogenesis were observed for mutant versus WT G1 mice (Fig 5B, D, and F). However, the G5 K82Δ mice had a significant increase in the number of tubules that were not stageable (e.g., because of absence or paucity of spermatids) (Fig 5C and F). Of the G5 K82Δ tubules that could be staged, there was a significant decrease in early-stage I–III tubules and an increase in stage IV–XII tubules (Fig 5E). Strikingly, we observed two distinct phenotypes in G5 K82Δ mice: a loss of the ordered organization of tubules and an elevation in the frequency of empty tubules (Fig 5G–I). Although these results highlight the deleterious effects of the K82Δ mutation in the mouse male reproductive system, they raise the question of whether the TPP1 mutation leads to loss of germ cells, somatic cells, or both cell types in the testis.

## K82Δ mutation results in a reduction of germline stem cells but not somatic cells

Within the seminiferous tubules, somatic cells known as Sertoli cells are essential "nurse" cells critical for germ cell development. To better understand if K82Δ elicits its effects directly through the germline compartment rather than through a broader impact on reproductive tissue development, we quantified the number of spermatogonia, spermatocytes, and Sertoli cells in WT and K82Δ G1 and G5 mice. Promyelocytic leukemia zinc-finger (PLZF) was used as a spermatogonial marker (Buaas et al, 2004; Costoya et al, 2004; Lovelace et al, 2016) to examine the effects of K82Δ on this germline compartment (Fig 6A). The total PLZF+ foci were counted per tubule in all tubules for each genotype in G1 and G5 mice. G1 WT and G1 K82Δ mice contained an average of 5.2 and 5.1 spermatogonia per tubule, respectively. Similarly, G5 WT tubules contained an average of 4.7 spermatogonia per tubule. However, G5 K82Δ mice

---

epididymis from G5 WT and G5 K82Δ mice. n = 4. *P* = 0.0067. **(H)** Representative sperm morphology from G5 WT and G5 K82Δ mice by H&E staining. Magnified views of boxed areas are shown on the right. Scale bar: 20 $\mu$m. **(I)** Quantitation of the percentage of abnormal sperm. n = 4.

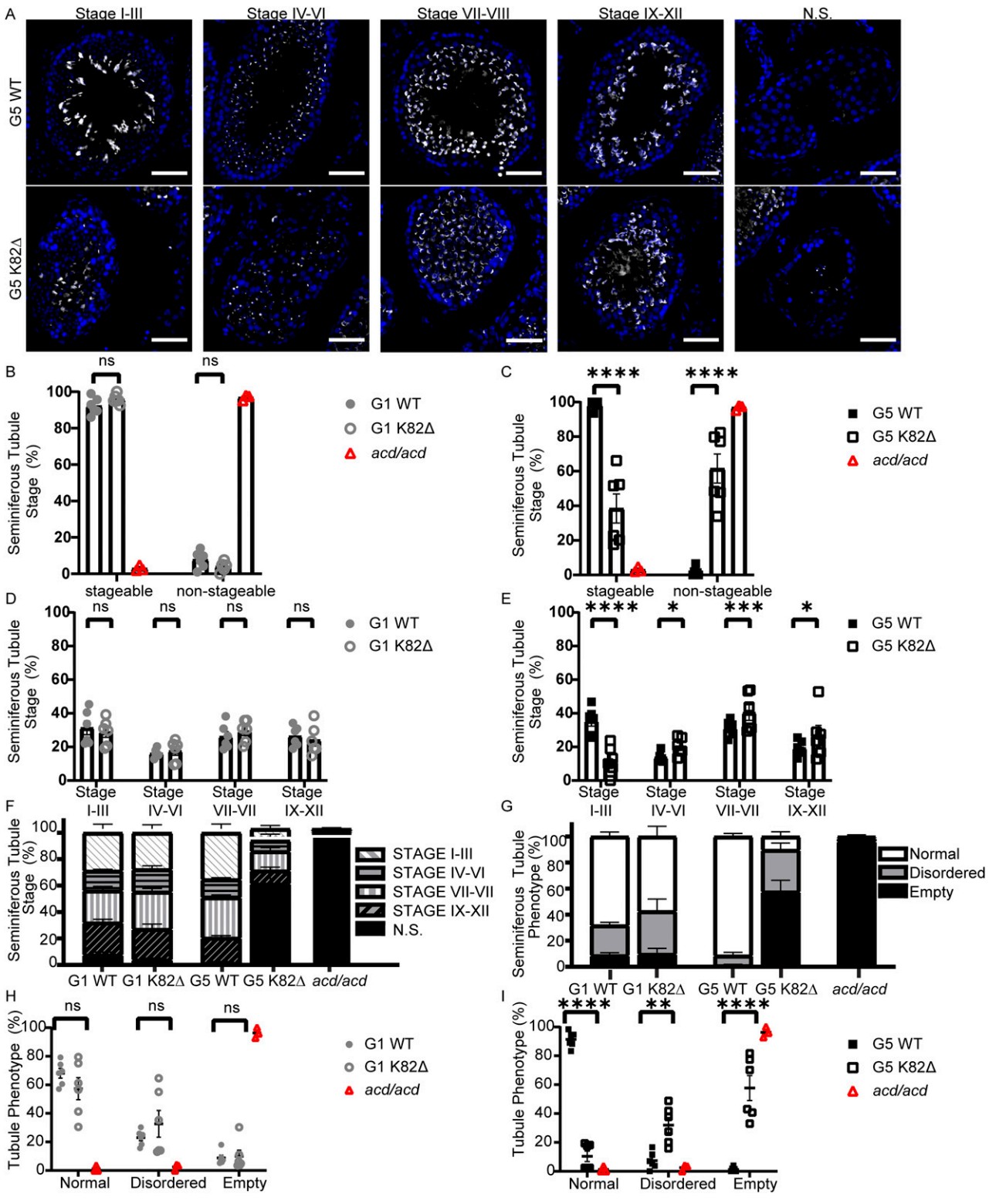

**Figure 5. Late-generation K82Δ mice testes are comprised primarily of disordered or empty tubules.**
**(A)** Immunofluorescence (IF) for PNA-lectin (spermatid acrosomes; grey) and DAPI (nuclei; blue) in cross sections of testes from G5 WT and G5 K82Δ mice in each stage including those that were not-stageable (N.S.) because they were devoid of spermatids, which are necessary for accurately staging tubules. Scale bar: 50 μm.
**(B, C)** Quantitation of total tubules that are stageable or not-stageable in (B) G1 WT, G1 K82Δ mice, and *acd/acd* or (C) G5 WT, G5 K82Δ, and *acd/acd* mice. **(D, E)** Quantitation of the percentage of stageable seminiferous tubules in a given stage of spermatogenesis in (D) G1 WT and G1 K82Δ mice and (E) G5 WT and G5 K82Δ mice. **(F)** Quantitation of the breakdown of each stage within G1 WT, G1 K82Δ, G5 WT, G5 K82Δ, and *acd/acd* mice. White diagonal lines indicate % of tubules in stages I–III, black horizontal lines

exhibited a marked reduction in the number of spermatogonia, averaging 2.8 spermatogonia per tubule (Fig 6B). This decrease cannot be attributed to an altered tubule size as G5 K82Δ testes displayed normal tubule diameter (Fig 4E). Meiotic prophase I spermatocytes were identified using SCP3 (Synaptonemal Complex Protein 3) (Fig 6C) (Yuan et al, 2000). G1 WT, G5 WT, and G1 K82Δ testes displayed a similar number of spermatocytes per tubule (averages of 65.0, 63.9, and 63.8, respectively). G5 K82Δ mice exhibited a significant reduction in the number of spermatocytes per tubule (average of 27.2) (Fig 6D). The decrease in both spermatogonia and spermatocytes was consistent with the apoptosis-mediated clearance of the male mouse germline described in $mTR^{-/-}$ models (Lee et al, 1998; Hemann et al, 2001), although in older (15–17-mo) G5 K82Δ mice that have already undergone substantial germline degeneration, we did not observe apoptotic markers likely because of prior clearance of dead cells (Fig S7B and D). Sertoli cells were identified using SOX9 as a marker (Fig 6E) (Chaboissier et al, 2004; Barrionuevo et al, 2006). In stark contrast to the observations in the germline, G5 K82Δ mice did not show a decrease in the number of Sertoli cells per tubule compared with G1 WT, G5 WT, or G1 K82Δ mice. Instead, a mild, but statistically significant, increase in the number of Sertoli cells per tubule was observed in G5 K82Δ compared with G5 WT control mice (Fig 6F). However, no changes in BrdU uptake seemed to contribute to this increase (Fig S7A and C). A decrease in both spermatogonia and spermatocytes but an increase in the number of Sertoli cells was also seen in $acd/acd$ hypomorphic mice, suggesting an inverse correlation between the phenotypes observed in the germline versus the somatic components of the mutant seminiferous tubules. Spermatogonial stem cells are highly proliferative and thus depend on telomerase to overcome the end replication problem (Pech et al, 2015). These data suggest that the reduction in telomere length maintenance because of reduced telomerase recruitment to telomeres by TPP1 K82Δ limits spermatogonia replicative lifespan, resulting in eventual germline stem cell exhaustion. As Sertoli cells of mature testes are largely non-proliferative, a lack of a severe phenotype in this compartment is not surprising. A modest increase in the number of Sertoli cells is consistent with activation of a compensatory mechanism in the face of germline failure, as Sertoli cells nourish developing sperm (Franca et al, 2016; Larose et al, 2019). Together, our data demonstrate that the TPP1 K82Δ mutation reduces mouse germline stem cell count and culminates in diminished gamete production that causes infertility in males.

# Discussion

### TPP1 K82Δ causes telomere shortening but not BM failure in mice

We report the first mouse model for telomerase-dependent telomere shortening caused by a telomeropathy mutation that leaves end protection and the structure and composition of telomerase intact. Homozygosity for this mutation results in telomere shortening that is discernible as early as in G1 and progresses with each successive generation. To our knowledge, our study also provides the most rigorous analysis of the HSC compartment in mice undergoing telomere shortening. We conducted a comprehensive analysis of the hematopoietic compartment spanning stem, progenitor, and mature cells of the BM, spleen, thymus, and blood. CBC analysis revealed age-related changes for both WT and K82Δ mice, but no consistent differences between the two genotypes were observed across time points in any of the hematopoietic compartments tested. Unlike in patients, but like mice knocked out for telomerase, K82Δ mice do not develop spontaneous BM failure. Based on other mouse models of telomerase deficiency (Lee et al, 1998; Herrera et al, 1999, 2000; Rudolph et al, 1999; Samper et al, 2002; Allsopp et al, 2003; Choudhury et al, 2007; Rossi et al, 2007; Armanios et al, 2009; Sekulovic et al, 2011), we would predict that K82Δ BM would become defective if exposed to severe stress caused by serial transplantations, or other types of hematopoietic stress. However, at least in steady state, these mice are resilient to BM failure. In conclusion, a TPP1 mutation that decreases telomere length in both humans and mice causes BM failure in humans but no deleterious changes at any stage of mouse hematopoiesis for up to five successive generations of breeding.

### Telomere shortening from TPP1 K82Δ results in decreased germline function in male mice

Infertility has been reported in $mTR^{-/-}$ mice (Lee et al, 1998), but reproducing this phenotype with a mutation in TPP1 that solely disrupts recruitment of, but not the expression or composition of, telomerase confirms beyond reasonable doubt that telomerase-dependent telomere shortening is sufficient to cause mouse infertility in the absence of concomitant spontaneous HSC failure. K82Δ mice started developing fertility defects at G4 that culminated in only one viable G6 (female) K82Δ mouse. Reduction in testes size, seminiferous tubule number, and sperm count were visible as early as G3 in K82Δ male mice. Furthermore, K82Δ sperm have a globozoospermia-like phenotype (round-shaped sperm head), which has been previously linked with male infertility (Yan, 2009). To our knowledge, we are the first to observe a split phenotype from telomere shortening as some K82Δ seminiferous tubules showed loss of germ cells, whereas others exhibited severe disorganization. It is unclear whether tubule disorganization and depletion of germ cells occur sequentially or if the two phenotypes are alternative outcomes of telomere shortening in the spermatogonial stem cells. Our in-depth quantitation of cell types in the testis confirmed that the TPP1 mutation decreases cell numbers for spermatogonia and differentiating cell types (spermatocytes and spermatids). Somatic cells in the testes such as Sertoli cells were completely spared by the K82Δ mutation, consistent with a germline-specific mechanism

are stages IV–VI, grey vertical lines are stages VII–VIII, black diagonal lines are stages IX–XII, and black filled bars are not-stageable. **(G)** Quantitation of tubule phenotype denoted as either normal organization (white bar), disordered organization (grey bar), or empty tubules (black bar) in G1 WT, G1 K82Δ, G5 WT, G5 K82Δ, and $acd/acd$ mice. **(H, I)** Quantitation of the percentage of seminiferous tubules with either normal, disordered, or empty tubule phenotype in (H) G1 WT, G1 K82Δ, and $acd/acd$ mice and (I) G5 WT, G5 K82Δ, and $acd/acd$ mice. $acd/acd$ was used as a positive control of gonadal defect. In panels (B, D, H): G1 WT (grey filled circles), G1 K82Δ (grey open circles). In panels (C, E, I): G5 WT (black filled squares), G5 K82Δ (black open squares). In panels (B, C, H, I): $acd/acd$ (red open triangle).

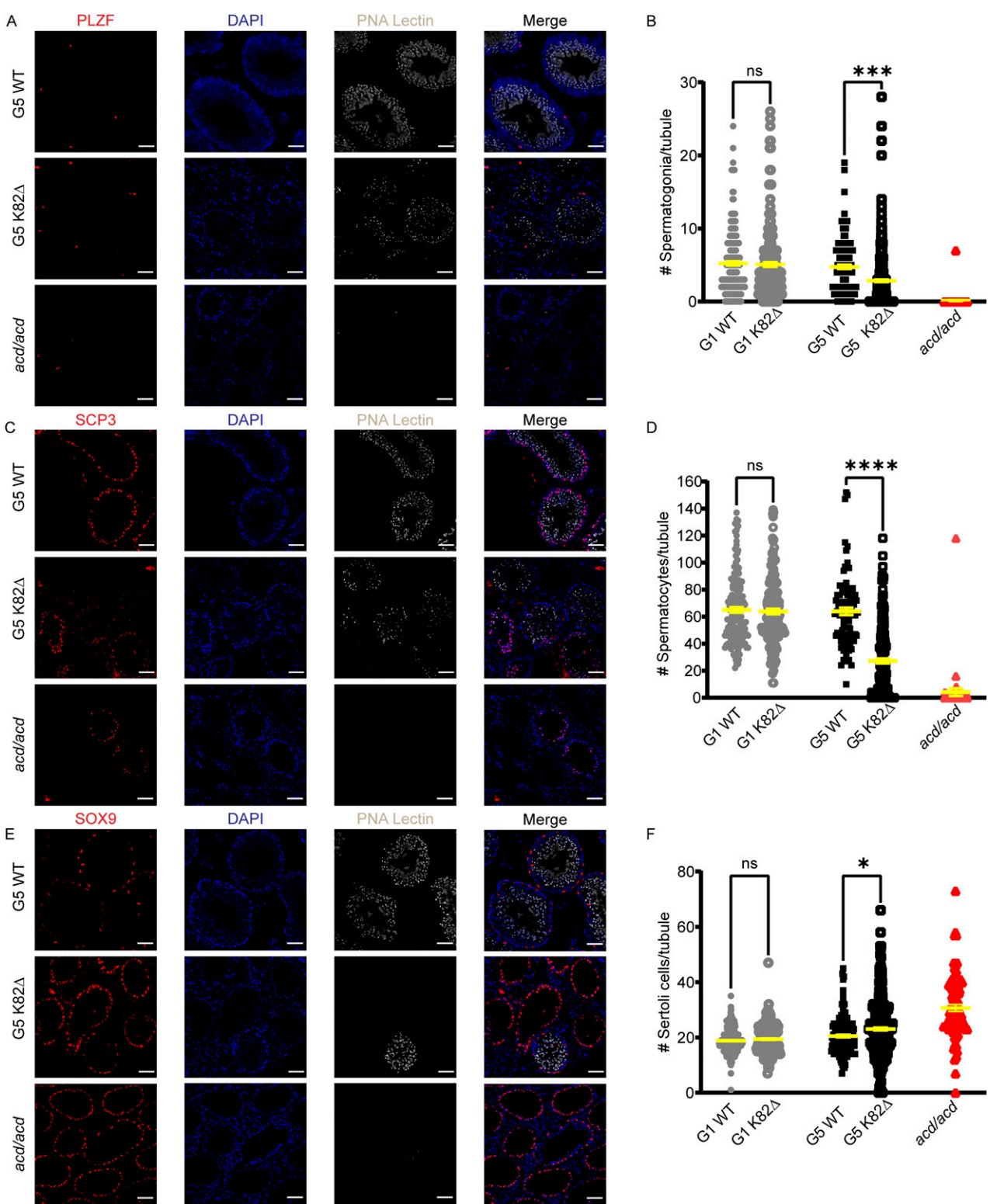

**Figure 6. K82Δ mutation results in a reduction of germ cells but an increase in somatic Sertoli cells.**
**(A, C, E)** Immunofluorescence for (A) PLZF (undifferentiated spermatogonia; red) (C) SCP3 (spermatocytes; red), (E) SOX9 (Sertoli cells; red), DAPI (nuclei; blue), and PNA Lectin (spermatid acrosomes; grey) in cross sections of testes from G5 WT, G5 K82Δ mice, and *acd/acd* mice. Scale bar: 50 μm. **(B, D, F)** Quantitation of the number of (B) spermatogonia, (D) spermatocytes, and (F) Sertoli cells per tubule in G1 WT, G1 K82Δ G5 WT, G5 K82Δ, and *acd/acd* mice. See the Materials and Methods section for total number of mice and tubules analyzed.

for the downstream infertility defect. In fact, K82Δ slightly increased the number of nursing Sertoli cells, perhaps to compensate for reduced germline function. It will be interesting to determine if germ cell loss results solely from stem cells succumbing to the end replication problem or if premature differentiation of spermatogonia as a result of telomere dysfunction also contributes to this phenotype. Although our study was restricted to the effect of TPP1 K82Δ on the mouse male germline (as early stages of mouse female germline development occur in utero and are therefore difficult to study), both male and female mice with the mutation were unable to produce offspring with wild-type mating partners. It should be noted that previous work in telomerase-deficient mouse models as well as this study show variance in the onset of phenotypes (G1 versus G3 versus G5) which could be due to differences in baseline telomere length at the very start of breeding. However, the sequence of defects is distinctly reproducible with the germline being affected first in steady-state conditions across mouse models. Together with previous studies, our data suggest that telomere lengthening afforded by telomerase is critical to the maintenance of the mouse germline. Although telomere attrition in reproductive tissues has been proposed to contribute to the natural process of reproductive aging in humans (Keefe, 1998; Kalmbach et al, 2013), the above defects we see in K82Δ mice have not been noted in telomeropathies. Thus, the mouse germline seems to be more vulnerable towards telomere shortening than the human germline.

### Implications for mammalian models of telomeropathies

Combined with previous findings in telomerase KO mice, our data mandate a change in the use of existing models to understand the importance of telomerase-mediated telomere lengthening in human health and disease. The resilience of mice to the somatic absence of telomerase-mediated telomere lengthening is well known. It has been hypothesized that the abnormally long telomere length in laboratory mice is primarily responsible for this species-specificity although it is also likely that these distinct vulnerabilities are, at least in part, manifestations of the different evolutionary strategies of mice and humans. Hematopoietic defects are observed in $mTR^{-/-}$ mice in a CAST/EiJ background that harbors short telomeres, qualifying them as a potential system to model telomeropathies (Armanios et al, 2009). However, it is unlikely that resetting telomeres to a shorter length is sufficient to reverse the long-optimized evolutionary strategy of mice. In this regard, it is likely that mouse strains with shorter telomeres also suffer from reduced reproductive capacity compared to the wild-type strain. Instead of "humanizing" mice, it seems more appropriate, although more challenging, to investigate the biomedical relevance of telomerase and telomere length maintenance in mammals that more closely mimic humans in their vulnerability to telomere shortening.

## Materials and Methods

### Oligonucleotides and Sanger DNA sequencing

All DNA oligonucleotides, including PCR primers, DNA coding for guide RNAs, and the ssODN repair templates were purchased from Integrated DNA Technologies. Sanger sequencing was performed in the Advanced Genomics Core at the University of Michigan.

### Mice

A CRISPR-Cas9 mouse line carrying a specific mutation in the *Acd/TPP1* gene (K82Δ; equivalent to human mutation K170Δ) in a C57BL/6 (B6) background was generated at the Transgenic Animal Model Core at the University of Michigan under the supervision of Thomas Saunders. Purified DNA/RNA was microinjected into fertilized eggs obtained by mating (C57BL/6 X SJL)F1 or C57BL/6 female mice with (C57BL/6 X SJL)F1 male mice and pronuclear microinjection was performed as described (Becker & Jerchow, 2011). A heterozygous founder was obtained using this approach and was genotyped using Sanger sequencing for the existence of the K82Δ mutation and the absence of any other unwanted changes in the flanking regions on the *Acd* locus. The founder mouse was crossed with WT B6 mouse to generate the line and backcrossed for four generations, thereby eliminating any off-target effects of genome editing. Once backcrossed, heterozygous mice were intercrossed to produce homozygous mice starting with generation 1 (G1). Homozygous mice were crossed together to breed successive generations (G2, G3, and so on). WT mice were also bred in parallel. All animals were housed in environmentally controlled conditions with 14 h light and 10 h dark cycles with food and water provided ad libitum. Mice were harvested at the indicated times for the generations tested in this study: G1: 23–25 mo, G3: 16–16.5 mo, G4: 14–21 mo, and G5: 15.5–17 mo. All protocols were approved by the Institutional Animal Care & Use Committee (IACUC) at the University of Michigan and the University of Pennsylvania and comply with policies, standards, and guidelines set by the States of Michigan and Pennsylvania and the United States Government.

### Screening for CRISPR-Cas9 editing and mutagenesis efficiency

In preliminary experiments leading up to the generation of CRISPR-Cas9 edited mice, the efficiency of *Acd* gene editing was evaluated. For this, blastocysts were harvested after injection with guide RNAs and donor oligonucleotide template. PCR reactions were conducted using blastocyst genomic DNA as a template and primers flanking the edited *Acd* locus. KpnI digestion was conducted with the PCR products to screen for the successful editing of the *Acd* locus.

### Telomere restriction fragment (TRF) analysis by Southern blot

Telomere length analysis was performed as described previously (Bisht et al, 2016) with a few modifications. Briefly, genomic DNA was isolated from harvested calf thymocytes using the GenElute kit (Sigma-Aldrich) after washing the cell pellets twice with PBS. Genomic DNA was similarly isolated from TPP1-S cells, which were used as a control (Grill et al, 2019). During DNA extraction, proteinase K treatment was conducted for 2 h 30 min at 55°C followed by incubation for 30 min at 70°C. 1 μg of genomic DNA was digested overnight with RsaI and HinfI at 37°C. DNA digests were run on a 0.7% agarose 1× tris-acetate-EDTA gel at 50 V for 4 h 30 min. The gel was imaged using EtBr staining with a fluorescent phospho-ruler aligned to wells and gel. Next, the gel was placed in denaturation

buffer for 20 min and then rinsed in water for 10 min. The gel was then placed on two sheets of Whatman 3MM filter paper for 1 h at RT. Once dried, the gel was placed in neutralizing solution for 15 min. After a water rinse, the gel was pre-hybridized in prewarmed church buffer for 20 min at 55°C. After pre-hybridization, 5′ $^{32}$P-labeled (TTAGGG)$_4$ oligonucleotide (labeled using [γ-$^{32}$P]ATP and T4 PNK; New England Biolabs) was added, and hybridization was continued overnight. After hybridization, the gel was washed thrice with 2× SSC followed by two more washes in 0.1× SSC/0.1% SDS at RT. The gel was transferred to filter paper, wrapped in plastic wrap, and exposed to a phosphor-imager screen, analyzed using the Imagequant TL software, and visualized on ImageJ. The gel was calibrated (Imagequant TL software) using the known molecular weights of the radiolabeled DNA ladder run on the same gel.

### Flow-FISH

This method uses FISH and flow cytometry to measure telomere length (Baerlocher et al, 2006). Single cell suspensions were prepared from BM followed by red blood cell lysis (R7757; Sigma-Aldrich). After resuspending cells with DMEM and 4% FBS, cells were counted. Each sample was split into four different centrifuge tubes with 1 × 10$^6$ cells per tube. Of these tubes, two were mixed with calf thymocytes (1 × 10$^6$ cells) of known telomere length (determined using TRF Southern blot analysis) as an internal control. One set of single and mixed samples was stained with a FITC PNA probe and the other set was left unstained to correct for auto-fluorescence (Dako telomere kit). DNA was denatured for 10 min at 82°C and allowed to hybridize overnight. The excess probe was removed using wash solution (Dako telomere kit) and heat (40°C for 10 min). After two rounds of washing, DNA was counterstained with LDS 751 for 5 h at 4°C (protected from light using aluminium foil). Cells were analyzed using a BD LSRFortessa flow cytometer and data were analyzed with FlowJo (Treestar/BD). Relative telomere length was calculated as a ratio of the difference of PNA signal in mouse BM samples with and without PNA probe and the difference of PNA signal in calf thymocytes in the same tube with and without PNA probe. Relative telomere length was then multiplied by 16.521 kb, which is the TRF length of calf thymocytes determined by Southern blot analysis, to obtain absolute telomere length.

### Flow cytometry

Single cell suspensions were prepared from spleen, BM, or thymus, followed by red blood cell lysis (for spleen and BM only) (R7757; Sigma-Aldrich). The following antibodies were from BioLegend: anti-CD4 (clone GK1.5), anti-CD8 (clone 53-6.7), anti-CD11b (clone M1/70), anti-CD19 (clone 6D5), anti-CD48 (clone HM48-1), anti-CD150 (clone TC15-12F12.2), anti-Ly6C/G (Gr-1) (clone RB6-8C5), anti-B220 (clone RA3-6B2), anti-Ly6A/E (Sca-1) (clone D7), anti-Ter119 (clone TER-119), anti-CD41 (clone MWReg30), anti-CD105 (clone MJ7/18), anti-CD16/32 (clone 93), anti-CD43 (clone S11), anti-IgM (clone AF6-78), anti-CD93 (clone AA4.1), anti-CD44 (clone IM7), anti-cKit/ CD117 (clone 2B8), and anti-Lineage, with an antibody cocktail containing the following: anti-CD3e (clone 17A2), CD4 (clone GK1.5), CD8 (clone 53-6.7), TCRβ (clone H57-597), TCRγδ (clone GL3), NK1.1 (clone PK136), CD11b (clone M1/70), CD11c (clone N418), Ter119 (clone

TER-119), Gr-1 (clone RB6-8C5), B220 (clone RA3-6B2), and CD19 (clone 6D5). Dead cells were excluded with Zombie Aqua Fixable Viability Dye (BioLegend). Cells were analyzed using a BD LSRFor-tessa flow cytometer (Becton Dickinson) and data were analyzed with FlowJo (TreeStar/BD).

### Complete blood counts

Blood was obtained through submental or submandibular bleeding and transferred to EDTA-treated tubes. CBCs were determined using the Advia 2120 (Siemens) and the Hemavet 950 veterinary analyzer (Drew Scientific).

### Histology

Mice were euthanized by isoflurane anesthesia followed by cervical dislocation. Testes were removed and fixed overnight in Bouin's solution or 4% paraformaldehyde. Samples were dehydrated through a graded series of ethanol, embedded in paraffin, and sectioned at 7 µm thickness with a Spencer Microtome (American Optical). Sections were stained with hematoxylin and eosin (H&E) following standard protocols. After staining, the slides were mounted with coverslips and Permount mounting media (Fisher SP15-500) and allowed to dry overnight in a fume hood. Digital images were captured using a Leica upright DM5000B microscope and Leica DFC310 FX Digital camera.

### Immunofluorescence (IF)

IF was performed as described previously (Vlangos et al, 2009). Briefly, deparaffinized sections were boiled in 0.1 M Sodium Citrate for 10 min for antigen retrieval and sections were blocked in suppressor serum (5% goat serum [#005-000-1210; Jackson ImmunoResearch], 95% blocking solution [3% BSA and 0.5% Tween in PBS]) for 20 min at RT in a humidifying chamber. Sections were incubated with primary antibodies diluted in suppressor serum overnight at 4°C in a humidifying chamber. At least one section for each slide was incubated in the absence of primary antibody to serve as a negative control. After a series of PBS washes, fluo-rophore-conjugated secondary antibodies diluted in suppressor serum were added to all sections in the dark and incubated in a humidifying chamber at RT for 1 h. Sections were washed again with PBS and counterstained with DAPI and PNA-lectin for 1 h at RT in a humidifying box. After the final three PBS wash steps, mounting media (Fisher SP15-500; Permount) was added and the coverslips and slides were left to dry overnight before imaging and stored at 4°C. Digital images were captured with the Olympus BX53F mi-croscope, Olympus DP80 digital camera, and CellSens Standard software. Antibodies and stains used are as follows: mouse anti-SCP3 (1:200, ab97672; Abcam), rabbit anti-PLZF (1:200, sc-22839; Santa Cruz Biotechnology), rabbit anti-Sox9 (1:666, AB5535; Millipore/Sigma-Aldrich), goat anti-rabbit IgG, Alexa Fluor 488 (1: 200, A11008; Invitrogen/Thermo Fisher Scientific), goat anti-rabbit IgG, Alexa Fluor 568 (1:200, A11011; Invitrogen/Thermo Fisher Sci-entific), goat anti-mouse IgG, Alexa Fluor 568 (1:200, A11004; Invitrogen/Thermo Fisher Scientific), goat anti-mouse IgG, Alexa Fluor 488 (1:200, A11001; Invitrogen/Thermo Fisher Scientific), Rat

anti-BrdU (1:200, OBT0030CX; Oxford Biotechnology), rabbit anti-cleaved caspase-3 (1:400, 9661S; Cell Signaling Technology), Lectin PNA, Alexa Fluor 488 conjugate (1:1,000; L21409; Invitrogen/Thermo Fisher Scientific), DAPI (1:1,000, 71-03-00; Kirkegaard & Perry Laboratories).

### Fertility measurement

Data were extracted from breeding cage cards to determine the litter sizes and the number of pups that survived past weaning for each mating pair of WT and K82Δ mice generations G1 through G5.

### Testes weight measurement

Mice were individually weighed before euthanasia, following which testes were dissected, weighed, and photographed. Quantitation of testes weight was made using testes weight per mouse total weight for each mouse.

### Sperm count and morphology

Sperm was collected by dissecting out vas deferens and epididymis, mincing into a single suspension with PBS, and incubation for 1 h at 37°C. Sperm were diluted, spotted on slides and allowed to dry. Six slides per mouse were analyzed (n = 4 G5 WT and n = 4 G5 K82Δ). Sperm were counted using a Makler chamber, with a minimum of four rows. Slides were then fixed in methanol for 15 min and stained with H&E as described above. Sperm morphology was quantified blinded and in triplicate. Sperm were designated as normal if they had an intact comma-shaped head, midpiece, and tail. Abnormal indicated a deviation from normal with the major difference being in head morphology.

### Staging of seminiferous tubules and evaluation of ordered/disordered tubules

Cell quantitation was performed to attain the absolute numbers of each cell type in this study. In each testis from G1 WT, G1 K82Δ, G5 WT, G5 K82Δ mice, and acd/acd, three 7 μm-thick sections, separated by a distance of more than 100 μm, were cut and analyzed. In all seminiferous tubules, the stages of spermatogenesis were determined by the shape of acrosomes stained by PNA-lectin. Staging was performed blinded. Number of mice and tubules analyzed: G1 WT: two mice, 532 tubules; G1 K82Δ: two mice, 501 tubules; G5 WT: two mice, 330 tubules; G5 K82Δ: two mice, 729 tubules; acd/acd hypomorph: one mouse, 175 tubules.

### Quantitation of germline and Sertoli cell number per tubule

Cell number counting was performed to attain the absolute numbers of each cell type. In each testis from G1 WT, G1 K82Δ, G5 WT, G5 K82Δ mice, or acd/acd hypomorph, at least two 7-μm-thick sections, separated by a distance of more than 100 μm, were cut and analyzed. Undifferentiated spermatogonia were identified using PLZF, spermatocytes were identified using SCP3, and Sertoli cells were marked with SOX9. Each cell positive for each marker was counted per seminiferous tubule for all tubules using ImageJ

software (National Institutes of Health [NIH]; http://imagej.nih.gov/ij/). Different sections from the same mouse were co-immunostained for the following pairs of markers: PLZF and BrdU, Cleaved Caspase-3 and SCP3, and BrdU and SOX9. After the addition of secondary Alexa fluor–conjugated antibodies, all sections were stained with PNA-lectin and DAPI and imaged at 200× total magnification using a 20× objective lens. In all sections, long (i.e., non-circular) tubules were excluded as they likely represent longitudinal rather than radial sections. Number of mice and tubules analyzed for each marker: PLZF/BrdU: G1 WT: two mice, 182 tubules; G1 K82Δ: two mice, 185 tubules; G5 WT: two mice, 116 tubules; G5 K82Δ: two mice, 264 tubules; acd/acd hypomorph: one mouse, 49 tubules. Caspase-3/SCP3: G1 WT: two mice, 161 tubules; G1 K82Δ: two mice, 158 tubules; G5 WT: two mice, 91 tubules; G5 K82Δ: two mice, 205 tubules; acd/acd hypomorph: one mouse, 41 tubules. BrdU/SOX9: G1 WT: two mice, 189 tubules; G1 K82Δ: two mice, 158 tubules; G5 WT: two mice, 123 tubules; G5 K82Δ: two mice, 260 tubules; acd/acd hypomorph: one mouse, 85 tubules.

### BrdU staining and quantitation

To detect cell proliferation, mice were injected intraperitoneally with BrdU at 100 mg/g body weight 24 h before organ removal. BrdU staining was performed as described in IF protocol above. BrdU positive tubules were designated positive if they had two or more positive foci. Images were scored blind to genotype. Quantitation of BrdU positive tubules in tubules that were stageable was performed and subjected to a t test to evaluate significant differences between WT and K82Δ in each generation. Number of mice and tubules analyzed: G1 WT: two mice, 371 tubules; G1 K82Δ: two mice, 343 tubules; G5 WT: two mice, 239 tubules; G5 K82Δ: two mice, 524 tubules.

### Cleaved Caspase-3 staining and quantitation

Cleaved Caspase-3 (CC3) IF was performed as described above. CC3 tubules were designated positive if tubules had at least one positive focus. Images were scored blind to genotype and the number of positive tubules was counted irrespective of the stage of spermiogenesis. A t test was calculated to determine significant differences between WT and K82Δ in each generation. Number of mice and tubules analyzed: G1 WT: two mice, 161 tubules analyzed; G1 K82Δ: two mice, 158 tubules; G5 WT: two mice, 91 tubules; G5 K82Δ: two mice, 205 tubules; acd/acd hypomorph: one mouse, 41 tubules.

### Statistical analysis

Statistical tests were performed using Prism software (GraphPad version 8). Experiments were analyzed between WT and mutant mice only. Analysis between generations was not included as they were not conducted at the identical age or cohort size. t tests were used between WT and mutant mice for each experiment. Graphs were generated in Prism software and represented as mean with 95% confidence interval for hematopoietic data and mean with SEM for germline data. Adjusted P-values for comparisons were reported as *$P < 0.05$, **$P < 0.01$, and ***$P < 0.001$. M or F symbols were superscripted to denote significance in either males (M) or females

(F). It should be noted that because of the many outcomes studied, there are some stochastic statistically significant results that are not consistent across genotype, generation, or sex. Independent confirmatory experiments would need to be conducted in a larger scale to follow up any of these isolated findings, especially those approaching $P = 0.05$.

## Data Availability

This study includes no data deposited in external repositories.

## Supplementary Information

## Acknowledgements

We thank Ashley Velez, Ritvija Agrawal, Benjamin Allen, Sunny Wong, Sherilyn Grill, Hande Kocak, members of Scott Leiser's laboratory (University of Michigan), and other members of Nandakumar, Maillard, and Keegan labs for intellectual discussions. We acknowledge Wanda Filipiak and Galina Gavrilina for preparation of transgenic mice and the Transgenic Animal Model Core of the University of Michigan's Biomedical Research Core Facilities and its Director, Thomas Saunders, for helping develop the CRISPR-based knock in strategy for generating TPP1 K82Δ mice. Transgenic Animal Model Core support was provided by The University of Michigan Rogel Cancer Center (supported by the National Cancer Institute of the National Institutes of Health under award number P30CA046592). We thank Marina Pasca di Magliano (University of Michigan) for kindly gifting the SOX9 antibody and use of Olympus BX53F microscope, Olympus DP80 digital camera, and CellSens Standard software. We are grateful to Jennifer Skidmore and Donna Martin for technical help with immunohistochemistry; Michael Pihalja at the University of Michigan Flow Cytometry Core for help with flow cytometry; Lindsay Moritz for help with sperm analysis; David Siemeniak, David Ginsburg, and Rami Khoriaty for help using instruments in their laboratory; and Gregg Sobocinski for help with microscopy. We thank the L & J Meat Market in Lake City, MI for donation of calf thymus for this study. This work was supported by National Institutes of Health (NIH) Grants R01AG050509 (CE Keegan, J Nandakumar, I Maillard), R01GM120094 (J Nandakumar), R01HD092084 (SS Hammoud), F30AI136315 (E Perkey), F30HD097961 (AN Shami), 5T32HD079342 (AN Shami), 5T32GM007863 (AN Shami), T32AG000114 (JV Graniel), T32HL07439 (LJ Carrington, JD Brandstadter), T32CA009140 (F Allen), and 1DP2HD091949-01 (SS Hammoud). The work was also supported by Open Philanthropy Grant 2019-199327 (5384) (SS Hammoud), Rackham Merit Fellowship from the University of Michigan (JV Graniel), and an American Cancer Society Research Scholar grant RSG-17-037-01-DMC (J Nandakumar).

## Author Contributions

JV Graniel: conceptualization, resources, data curation, formal analysis, supervision, funding acquisition, validation, investigation, visualization, methodology, project administration, and writing—original draft, review, and editing.
K Bisht: conceptualization and investigation.
A Friedman: investigation.
J White: investigation.

E Perkey: supervision, funding acquisition, investigation, and writing—review and editing.
A Vanderbeck: investigation and writing—review and editing.
A Moroz: investigation.
LJ Carrington: funding acquisition, investigation, and writing—review and editing.
JD Brandstadter: funding acquisition, investigation, and writing—review and editing.
F Allen: funding acquisition, investigation, and writing—review and editing.
AN Shami: supervision, investigation, methodology, and writing—review and editing.
P Thomas: investigation.
A Crayton: investigation.
M Manzor: investigation.
A Mychalowych: investigation.
J Chase: investigation.
SS Hammoud: resources, supervision, funding acquisition, and writing—review and editing.
CE Keegan: conceptualization, resources, supervision, funding acquisition, investigation, visualization, project administration, and writing—review and editing.
I Maillard: conceptualization, resources, supervision, funding acquisition, project administration, and writing—original draft, review, and editing.
J Nandakumar: conceptualization, resources, supervision, funding acquisition, methodology, project administration, and writing—original draft, review, and editing.

### Conflict of Interest Statement

The authors declare that they have no conflict of interest.

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
