## [Reviewer comments · Life Science Alliance]

Life Science Alliance

Differential impact of a dyskeratosis congenita mutation in TPP1 on mouse hematopoiesis and germline

Jacqueline Graniel, Kamlesh Bisht, Ann Friedman, James White, Eric Perkey, Ashley Vanderbeck, Alina Moroz, Léolène Carrington, Joshua Brandstadter, Frederick Allen, Adrienne Shami, Peedikayil Thomas, Aniela Crayton, Mariel Manzor, Anna Mychalowych, Jennifer Chase, Saher Hammoud, Catherine Keegan, Ivan Maillard, and Jayakrishnan Nandakumar

DOI: <https://doi.org/10.26508/lsa.202101208>

Corresponding author(s): Jayakrishnan Nandakumar, University of Michigan-Ann Arbor, Ivan Maillard, Raymond and Ruth Perelman School of Medicine at the University of Pennsylvania and Catherine Keegan, University of Michigan-Ann Arbor

Review Timeline:	Submission Date:	2021-08-23
	Editorial Decision:	2021-09-13
	Revision Received:	2021-09-26
	Editorial Decision:	2021-09-28
	Revision Received:	2021-09-30
	Accepted:	2021-10-01

Transaction Report:

September 13, 2021

Re: Life Science Alliance manuscript #LSA-2021-01208-T

Dr. Jayakrishnan Nandakumar
University of Michigan-Ann Arbor
MCDB
1105 N. University avenue
Ann Arbor, MI 48109

Dear Dr. Nandakumar,

Thank you for submitting your manuscript entitled "Differential impact of a dyskeratosis congenita mutation in TPP1 on mouse hematopoiesis and germline" to Life Science Alliance. The manuscript was assessed by expert reviewers, whose comments are appended to this letter. We invite you to submit a revised manuscript addressing the Reviewer comments.

Thank you for this interesting contribution to Life Science Alliance. We are looking forward to receiving your revised manuscript.

Sincerely,

Eric Sawey, PhD
Executive Editor
Life Science Alliance
<http://www.lsa-journal.org>

- A letter addressing the reviewers' comments point by point.
- An editable version of the final text (.DOC or .DOCX) is needed for copyediting (no PDFs).
- High-resolution figure, supplementary figure and video files uploaded as individual files: See our detailed guidelines for preparing your production-ready images, <https://www.life-science-alliance.org/authors>
- Summary blurb (enter in submission system): A short text summarizing in a single sentence the study (max. 200 characters including spaces). This text is used in conjunction with the titles of papers, hence should be informative and complementary to the title and running title. It should describe the context and significance of the findings for a general readership; it should be written in the present tense and refer to the work in the third person. Author names should not be mentioned.
- By submitting a revision, you attest that you are aware of our payment policies found here: <https://www.life-science-alliance.org/copyright-license-fee>

B. MANUSCRIPT ORGANIZATION AND FORMATTING:

Reviewer #1 (Comments to the Authors (Required)):

In this manuscript, Graniel and colleagues report on the impact of a dyskeratosis congenita mutation in TPP1 (K170Δ) using a novel mouse model. The authors generated a mutant mouse line harboring the equivalent mutation in TPP1 (K82Δ). Mice carrying the TPP1 K82Δ mutation show progressive telomere shortening confirming the impact of this mutation on telomere length regulation. Strikingly, the progressive telomere shortening observed in these mice does not result in hematopoietic defects, a hallmark of DC in humans, and a feature of telomerase deficient mice. However, late generation TPP1 K82Δ mice show mouse infertility, caused by severe defects in the testis, affecting spermatogenesis. Based on a comprehensive and rigorous analysis of this novel mouse model the authors propose that telomere length maintenance is crucial in the mouse germline to ensure multiple offspring. In contrast, in long-lived animals like humans, telomere length is essential to maintain vital tissues such as the bone marrow to ensure progression to reproductive

age.

Overall, this is a well-performed study that helps explain some of the differences in phenotypes induced by loss of telomere homeostasis between humans and mice. The generation of a novel mouse model for a DC-linked mutation will be of interest to the field. The experiments are well presented, and the data are convincing and well-controlled.

Reviewer #2 (Comments to the Authors (Required)):

In "Differential impact of a dyskeratosis congenita mutation in TPP1 on mouse hematopoiesis and germline" Graniel et al describe their studies on the hematopoietic and germline tissues of a mouse harboring a clinically relevant mutation in TPP1. The authors show no hematopoietic deficit, but severe germline impairment with telomere shortening. While most of the phenotypes described here have been described before in different mouse models of telomere deficit, the experiments are well performed and the mouse described extremely interesting. I have very few comments.

The authors try to justify (in their Introduction) the relevance of their study with what they call "controversial non-telomeric functions" of the telomerase holoenzyme. This is irrelevant and unnecessary. The phenotypes in DC have been linked many times to telomere dysfunction. Rescue experiments have been performed many times. This paper does not need to invoke controversies with non-canonical roles of telomerase to justify its relevance. These sentences should be removed.

The authors claim, both in their Abstract and in their Introduction, that their "studies support a model wherein the mouse germline and the human BM are especially vulnerable towards defects in telomerase-dependent telomere length maintenance". However, the authors didn't work with human BM samples at all, so clearly, their studies don't support this model. I agree that extensive literature agrees with this model, so these sentences should be rephrased.

The results obtained in the hematopoietic system (basically no signs of hematopoietic impairment), combined with the lack of skin aberrations (data not shown) is in line with what has been observed in previous studies with mTR^{-/-} mice. It represents a complete lack of DC-related phenotypes, indicating that lab mice are not a suitable model for the study of telomere-diseases, and therefore not an optimal model to develop novel alternatives to treatment of these patients. This should be mentioned in the text.

The results obtained in germline are very interesting and complete. As the authors mention, a lot of these phenotypes, or at least the main message, have been described before (Pech et al, 2015, and mTR^{-/-} mice papers).

The mouse model created and the experiments described here were rigorously performed. I do not have any comments on that. It is unfortunate that most of the data presented mimics what has been shown before in other telomere-maintenance deficient mice, but still due to the novelty of the mutation studied, I believe this paper warrants publication. However, I believe the manuscript has to be shortened, specifically the Discussion section. Unnecessarily long, reads more as a PhD thesis than as a manuscript. The entire section on the Evolutionary Model for Differential Vulnerabilities, while interesting, is more suited for a review than for a Discussion section of a paper (as no data was presented to actually support these interesting ideas)

We thank the Reviewers for their critiques. No requests were made for additional experiments. We have addressed all requested changes to the manuscript text in the revised manuscript. Please find below our point-by-point responses (starting with >>>) to each of the Reviewers' comments (*italicized and in green*).

Reviewer #1 (Comments to the Authors (Required)):

In this manuscript, Graniel and colleagues report on the impact of a dyskeratosis congenita mutation in TPP1 (K170Δ) using a novel mouse model. The authors generated a mutant mouse line harboring the equivalent mutation in TPP1 (K82Δ). Mice carrying the TPP1 K82Δ mutation show progressive telomere shortening confirming the impact of this mutation on telomere length regulation. Strikingly, the progressive telomere shortening observed in these mice does not result in hematopoietic defects, a hallmark of DC in humans, and a feature of telomerase deficient mice. However, late generation TPP1 K82Δ mice show mouse infertility, caused by severe defects in the testis, affecting spermatogenesis. Based on a comprehensive and rigorous analysis of this novel mouse model the authors propose that telomere length maintenance is crucial in the mouse germline to ensure multiple offspring. In contrast, in long-lived animals like humans, telomere length is essential to maintain vital tissues such as the bone marrow to ensure progression to reproductive age.

Overall, this is a well-performed study that helps explain some of the differences in phenotypes induced by loss of telomere homeostasis between humans and mice. The generation of a novel mouse model for a DC-linked mutation will be of interest to the field. The experiments are well presented, and the data are convincing and well-controlled.

>>>We thank the Reviewer for their comments. This Reviewer did not request any changes to the manuscript.

Reviewer #2 (Comments to the Authors (Required)):

In "Differential impact of a dyskeratosis congenita mutation in TPP1 on mouse hematopoiesis and germline" Graniel et al describe their studies on the hematopoietic and germline tissues of a mouse harboring a clinically relevant mutation in TPP1. The authors show no hematopoietic deficit, but severe germline impairment with telomere shortening. While most of the phenotypes described here have been described before in different mouse models of telomere deficit, the experiments are well performed and the mouse described extremely interesting. I have very few comments.

The authors try to justify (in their Introduction) the relevance of their study with what they call "controversial non-telomeric functions" of the telomerase holoenzyme. This is irrelevant and unnecessary. The phenotypes in DC have been linked many times to telomere dysfunction. Rescue experiments have been performed many times. This paper does not need to invoke controversies with non-canonical roles of telomerase to justify its relevance. These sentences should be removed.

>>>We agree with the Reviewer and have removed the mention of non-canonical roles of telomerase from the Introduction.

The authors claim, both in their Abstract and in their Introduction, that their "studies support a model wherein the mouse germline and the human BM are especially vulnerable towards defects in telomerase-dependent telomere length maintenance". However, the authors didn't work with human BM samples at all, so clearly, their studies don't support this model. I agree that extensive literature agrees with this model, so these sentences should be rephrased.

>>>We have rephrased sentences in the Abstract and Introduction to clarify that our results are only with the mouse model and that statements regarding human BM are inferred from past studies on DC.

The results obtained in the hematopoietic system (basically no signs of hematopoietic impairment), combined with the lack of skin aberrations (data not shown) is in line with what has been observed in previous studies with mTR^{-/-} mice. It represents a complete lack of DC-related phenotypes, indicating that lab mice are not a suitable model for the study of telomere-diseases, and therefore not an optimal model to develop novel alternatives to treatment of these patients. This should be mentioned in the text.

>>>We have included a section at the end of our Discussion describing how mice are not a suitable model for the study of telomere-shortening associated disease models.

The results obtained in germline are very interesting and complete. As the authors mention, a lot of these phenotypes, or at least the main message, have been described before (Pech et al, 2015, and mTR^{-/-} mice papers).

The mouse model created and the experiments described here were rigorously performed. I do not have any comments on that. It is unfortunate that most of the data presented mimics what has been shown before in other telomere-maintenance deficient mice, but still due to the novelty of the mutation studied, I believe this paper warrants publication. However, I believe the manuscript has to be shortened, specifically the Discussion section. Unnecessarily long, reads more as a PhD thesis than as a manuscript. The entire section on the Evolutionary Model for Differential Vulnerabilities, while interesting, is more suited for a review than for a Discussion section of a paper (as no data was presented to actually support these interesting ideas)

>>>We thank the Reviewer for their supportive comments on the germline work in this manuscript and the ideas surrounding the evolutionary basis for mouse versus human vulnerabilities towards telomere shortening. In accordance with the Reviewer's request, we have shortened all sections of the manuscript, especially the Discussion section, to keep our focus on the findings and immediate implications of the current study.

September 28, 2021

RE: Life Science Alliance Manuscript #LSA-2021-01208-TR

Dr. Jayakrishnan Nandakumar
University of Michigan-Ann Arbor
MCDB
1105 N. University Avenue
Ann Arbor, MI 48109

Dear Dr. Nandakumar,

Thank you for submitting your revised manuscript entitled "Differential impact of a dyskeratosis congenita mutation in TPP1 on mouse hematopoiesis and germline". We would be happy to publish your paper in Life Science Alliance pending final revisions necessary to meet our formatting guidelines.

- please add ORCID ID for secondary corresponding author-they should have received instructions on how to do so
- please use the [10 author names, et al.] format in your references (i.e. limit the author names to the first 10)
- please add callouts for Figures S1A-I; S3A-O; S4A-G; S5A0-G; S6A-E to your main manuscript text
- please add a scale bar to Figure S6E, and indicate its size in the legend

A. FINAL FILES:

B. MANUSCRIPT ORGANIZATION AND FORMATTING:

Sincerely,

October 1, 2021

RE: Life Science Alliance Manuscript #LSA-2021-01208-TRR

Dr. Jayakrishnan Nandakumar
University of Michigan-Ann Arbor
MCDB
1105 N. University avenue
Ann Arbor, MI 48109

Dear Dr. Nandakumar,

Thank you for submitting your Research Article entitled "Differential impact of a dyskeratosis congenita mutation in TPP1 on mouse hematopoiesis and germline". It is a pleasure to let you know that your manuscript is now accepted for publication in Life Science Alliance. Congratulations on this interesting work.

DISTRIBUTION OF MATERIALS:

Again, congratulations on a very nice paper. I hope you found the review process to be constructive and are pleased with how the manuscript was handled editorially. We look forward to future exciting submissions from your lab.

Sincerely,
